# Controlling protein function by fine-tuning conformational flexibility

Sonja Schmid[1†]*, Thorsten Hugel[1,2]*

[1]Institute of Physical Chemistry, University of Freiburg, Freiburg, Germany; [2]Signalling research centers BIOSS and CIBSS, Albert Ludwigs University, Freiburg, Germany

**Abstract** In a living cell, protein function is regulated in several ways, including post-translational modifications (PTMs), protein-protein interaction, or by the global environment (e.g. crowding or phase separation). While site-specific PTMs act very locally on the protein, specific protein interactions typically affect larger (sub-)domains, and global changes affect the whole protein non-specifically. Herein, we directly observe protein regulation under three different degrees of localization, and present the effects on the Hsp90 chaperone system at the levels of conformational steady states, kinetics and protein function. Interestingly using single-molecule FRET, we find that similar functional and conformational steady states are caused by completely different underlying kinetics. We disentangle specific and non-specific effects that control Hsp90's ATPase function, which has remained a puzzle up to now. Lastly, we introduce a new mechanistic concept: functional stimulation through conformational confinement. Our results demonstrate how cellular protein regulation works by fine-tuning the conformational state space of proteins.

*For correspondence:
thorsten.hugel@physchem.uni-freiburg.de (TH);
s.schmid@tudelft.nl (SS)

Present address: †Department of Bionanoscience, Kavli Institute of Nanoscience Delft, Delft University of Technology, Delft, Netherlands

Competing interests: The authors declare that no competing interests exist.

## Introduction

Protein function is essential for life as we know it. It is largely encoded in a protein's amino-acid chain that dictates not only the specific 3D structure, but also the conformational flexibility and dynamics of a protein in a given environment. Precise regulation of protein function is vital for every living cell to cope with an ever-changing environment, and occurs on many levels pre- and post-translationally (*Vogel and Marcotte, 2012*; *Hirano et al., 2016*). After translation by the ribosome, protein function depends strongly on post-translational modifications (PTMs) (*Deribe et al., 2010*), but also on binding of nucleotides (*Hodge and Ridley, 2016*), cofactors (*Weikum et al., 2017*), various protein-protein interactions (PPIs) (*Scott et al., 2016*), and global effects, such as temperature (*Danielsson et al., 2015*), macro-molecular crowding and phase separation (*Uversky, 2017*), redox conditions (*Levin et al., 2017*), osmolarity (*de Nadal and Posas, 2015*) etc. Importantly, this regulation occurs on very diverse levels of localization. Global effects affect the whole protein non-specifically, PPIs act at a given interface and site-specific modifications are very localized. Nevertheless, all of them influence the molecular properties that determine the 3D conformation, the conformational dynamics, and thereby also the function of a protein (*Schwenkert et al., 2014*; *Lewandowski et al., 2015*; *Schummel et al., 2016*; *Wei et al., 2016*; *Csizmok and Forman-Kay, 2018*). The chaperone protein Hsp90 (*Schopf et al., 2017*) is an excellent test system to investigate diverse regulation mechanisms (*Stetz et al., 2018*). It was recently discussed that a single PTM can functionally mimic a specific co-chaperone interaction in human Hsp90 (*Zuehlke et al., 2017*). Here we take a next step and disentangle how a PTM-related point mutation, a co-chaperone interaction, and macro-molecular crowding affect the function, kinetics, and thermodynamics of this multi-domain protein.

Hsp90 is an important metabolic hub. Assisted by about twenty known cochaperones, yeast Hsp90 is involved in the maturation of 20% of the entire proteome (*Taipale et al., 2010*). Among its substrates (referred to as 'clients') are many kinases involved in signal transduction, hormone

**eLife digest** Proteins play a wide variety of roles in the cell and interact with many other molecules. The behavior of proteins depends on their structure; yet, proteins are often flexible and will change shape, much like a tree in the wind. Nevertheless, for some of the activities that it performs, a protein must adopt one specific shape. Therefore, the likelihood that the protein will take on this specific shape directly determines how efficiently that protein can perform a specific job.

The shape of a protein can be regulated by changes at several levels; these could include modifying one of the amino acid building blocks that make up that protein, binding to another protein, or by placing the protein in a part of the cell that is crowded with other large molecules. Schmid and Hugel wanted to understand how these three different types of regulation affect the structure of a protein and how they relate to its activities.

The protein Hsp90 was used as a test case. It typically exists with two copies of the protein bound together, either in a parallel or a V-shape. Hsp90 plays several important roles in metabolism and can break down molecules of ATP, the so-called energy currency of the cell. All three types of regulation favored the Hsp90 pairs taking the parallel structure and increased its breakdown of ATP. The results suggest that the Hsp90 pair has a flexible structure, and that reducing this flexibility can improve Hsp90's efficiency in carrying out its role.

It was particularly unexpected that the large-scale, unspecific effect of placing the protein in a crowded environment could have such similar results to a small-scale, precise change of a single amino acid within the protein. While all three forms of regulation help to stabilize the parallel structure for Hsp90, they do this through different mechanisms, which influence the speed and the way that the protein transitions between the two structures. Schmid and Hugel believe that these results offer a new perspective on how diversely the shape and function of proteins is controlled at the molecular level, which could have wider implications for medical diagnostics and treatment.

receptors, the 'guardian of the genome p53' (*Lane, 1992*), and also cytoskeletal proteins such as actin, tubulin, and many more (*Pearl and Prodromou, 2006*; *Khandelwal et al., 2016*). Cancer cells were found to be 'addicted' to Hsp90 (*Trepel et al., 2010*), which is therefor also a prominent drug target in cancer research. Hsp90 is a homo-dimer where each monomer consists of three domains (*Ali et al., 2006*): the N-terminal domain (N) with a slow ATPase function, the middle domain (M) believed to be the primary client interaction site (*Verba et al., 2016*), and the C-terminal domain providing the main dimerization contacts. Apart from closed conformations, where the three domains align in parallel, Hsp90 exists primarily in V-shaped, open conformations with dissociated N-terminal and middle domains (*Krukenberg et al., 2008*; *Hellenkamp et al., 2017*). Both global arrangements are semi-stable at room temperature. As a consequence, Hsp90 alternates constantly between open and closed conformations - even in the absence of the chemical energy source, ATP (*Mickler et al., 2009*; *Schmid et al., 2016*; *Lee et al., 2019*). Surprisingly, the characteristic conformational changes of Hsp90 are only little affected by e.g. anti-cancer drug candidates (*Schmid et al., 2018*) or natural nucleotides (*Schmid et al., 2016*). In addition, to the stress-induced isoform discussed herein (Hsp82), there is also a cognate isoform (Hsc82) in yeast, which differs in unfolding stability, client range and more, despite 97% sequence identity (*Girstmair et al., 2019*).

Here, we present three orthogonal ways to modulate Hsp90's conformational state space, illustrated in *Figure 1*. The investigated point mutation A577I is located in the C-terminal hinge region of Hsp90. Residue A577 is the equivalent of the post-translational S-nitrosylation site C598 in human Hsp90 (*Martínez-Ruiz et al., 2005*). While nitrosylation of that residue has a two-fold inhibitory effect – on the ATPase function and the client stimulation by human Hsp90 – the A577I mutation caused a nearly 4-fold amplification of the ATPase rate (*Retzlaff et al., 2009*). The fact that the point mutation is located in the C-terminal domain and the ATP binding site in the N-domain indicates a long-range communication, offering valuable mechanistic insight in Hsp90's intra-molecular plasticity. Second, we consider the protein-protein interactions between Hsp90 and the activating co-chaperone Aha1, which is a well-known stimulator of Hsp90's inherently slow ATPase activity. It makes contacts to the middle and N-terminal domain, which rearranges the ATP lid (*Schulze et al., 2016*),

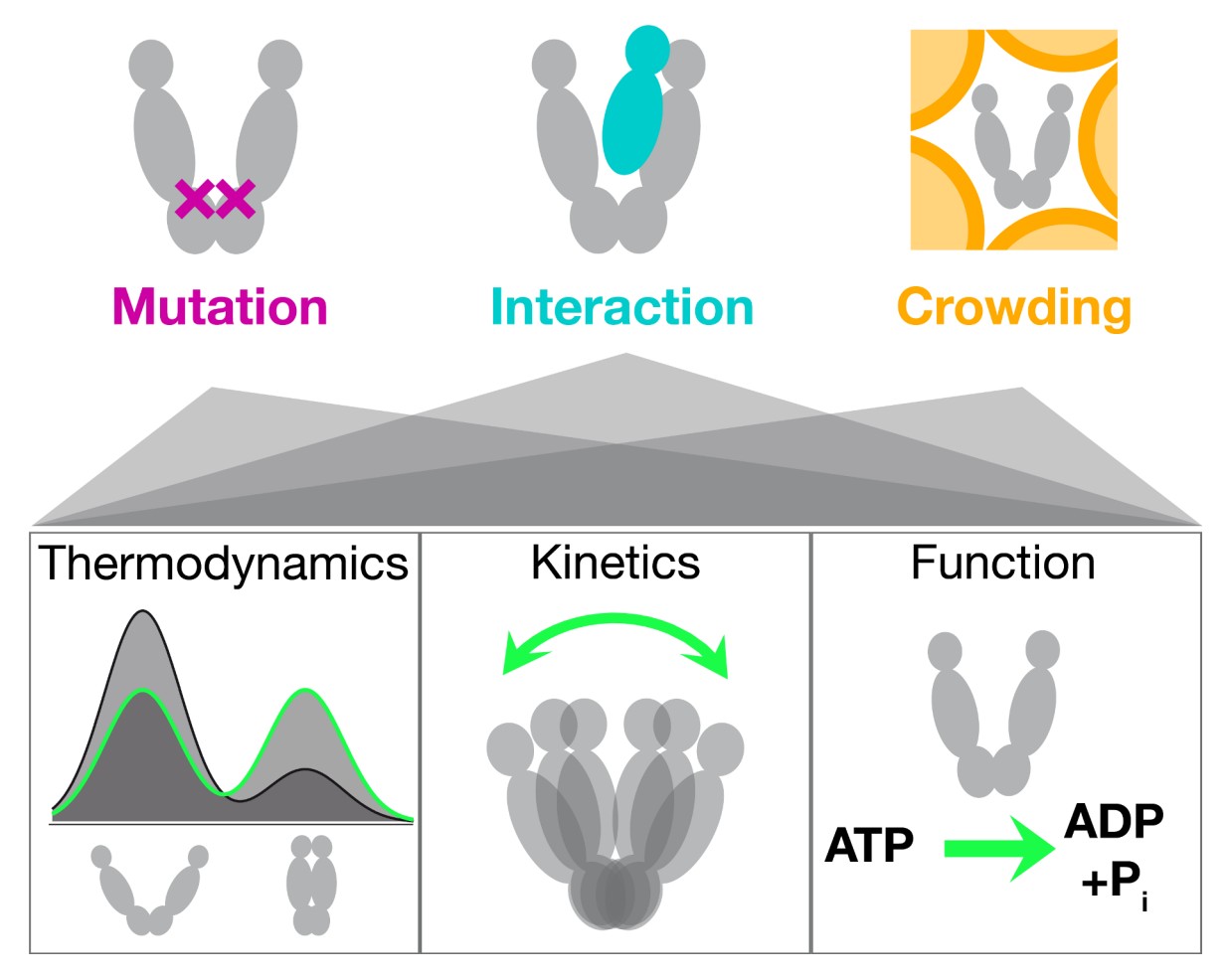

**Figure 1.** Protein regulation uses different degrees of localization. Mutations or PTMs act most locally, protein-protein interactions (PPIs) act on the protein domain level, and changes in the global environment, such as crowding or phase separation, act non-specifically and globally on the protein. Each of them affects conformational thermodynamics and kinetics to fine-tune the protein conformational state space and thereby protein function.

and the catalytic loop (including R380) (*Pearl, 2016*) in a favorable way for ATP hydrolysis. The affinity of Aha1 for Hsp90 itself is also markedly enhanced by PTMs (*Xu et al., 2019*). The third way of modulation mimics the crowding encountered in the cell, which is full of proteins, nucleic acids, vesicles and organelles. We mimic cellular macro-molecular crowding using the common crowding agent Ficoll400, that is branched polymeric sucrose. In contrast to the previous two modulations, crowding represents a completely non-specific, physical interference.

At first sight, all three modulations provoke a similar steady-state behavior in Hsp90. But our single-molecule experiments allow us to disentangle the different underlying causes thereof.

## Results

### Mutation, cochaperone and crowding show similar thermodynamics

First we follow yeast Hsp90's conformational kinetics in real-time using single-molecule Förster resonance energy transfer (smFRET) measured on a total-internal reflection fluorescence (TIRF) microscope (*Figure 2a*). The FRET pair configuration displayed in *Figure 2a* (top) results in low FRET efficiency (little acceptor fluorescence) for V-shaped, open conformations of Hsp90, and high FRET efficiency (intense acceptor fluorescence) for more compact, closed conformations. This allows us to obtain steady-state information, like the population of closed conformations, and also the kinetics of

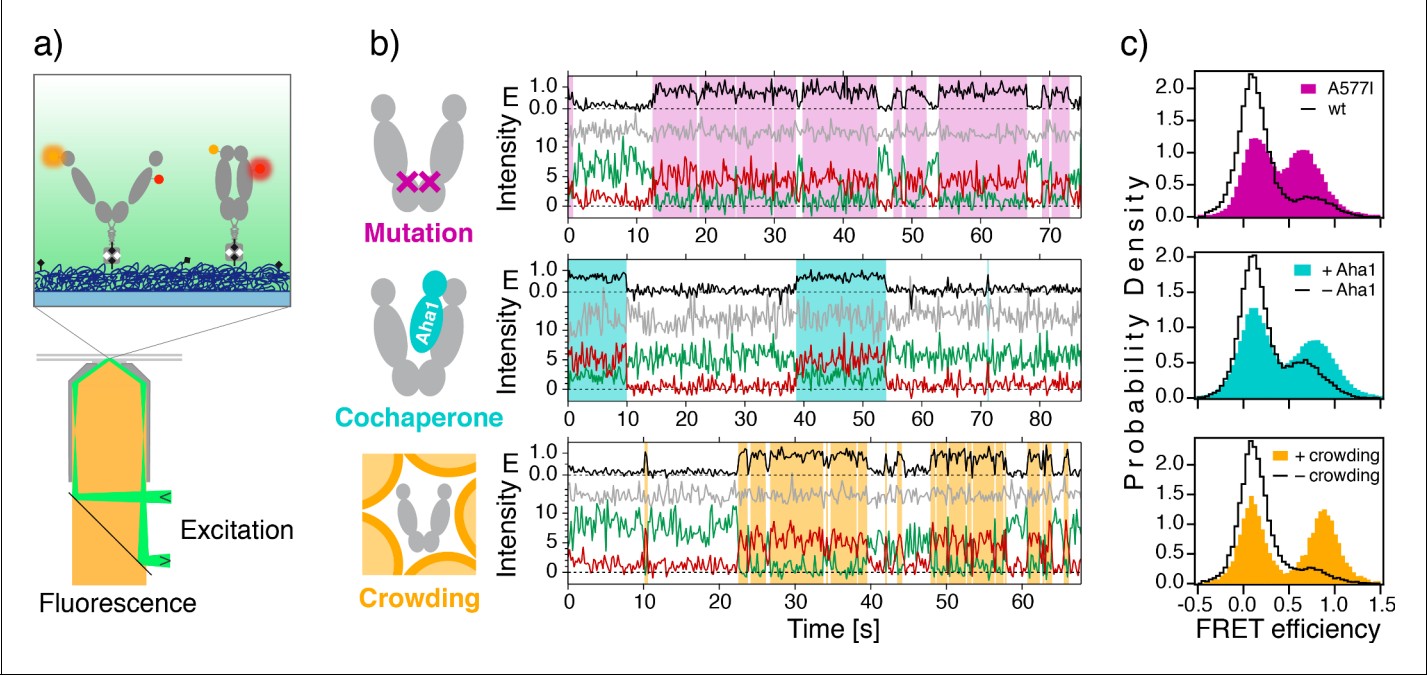

**Figure 2.** Mutation, cochaperone interaction, and crowding show similar thermodynamic effects. (a) Illustration of the single-molecule FRET experiment using an objective-type TIRF microscope (bottom): cross section through the objective, and flow chamber (both gray) and the dichroic mirror (black) separating the laser excitation (green) from the collected fluorescence (yellow). The zoom view (top) shows the fluorescently labeled Hsp90 (FRET donor, orange; acceptor red), which is immobilized on a PEG-passivated (dark blue) coverslip (light blue) using biotin-neutravidin coupling (black and gray). (b) Example time traces obtained from individual Hsp90 molecules for the point mutant A577I (top), in the presence of 3.5 µM cochaperone Aha1 (center), or under macro-molecular crowding by 20wt% Ficoll400 (bottom). Depicted are the FRET efficiency E (black), the fluorescence of the FRET donor (green) and acceptor (red) and the directly excited acceptor (gray). White and colored overlays denote low- and high-FRET dwells, respectively, as obtained using a hidden Markov model and the Viterbi algorithm. (c) FRET histograms compiled from many single-molecule trajectories as indicated, and normalized to unity (wt: wild type). Reference data (black) were measured under the specific conditions of each of the three experiment series (see also Materials and methods). Example traces of the reference data are provided in *Figure 2—figure supplement 1*. All fits and fit coefficients for 2 c are shown in *Figure 2—figure supplement 2*. Experimental conditions for A577I or wt Hsp90: 2 mM ATP in 40 mM Hepes, 150 mM KCl, 10 mM MgCl₂, pH7.5. Conditions with and without 3.5 µM Aha1: wt Hsp90, 2 mM ATP in 40 mM Hepes, 20 mM KCl, 5 mM MgCl₂, pH 7.5. Conditions with and without crowding by 20wt% Ficoll400: wt Hsp90 in 40 mM Hepes, 150 mM KCl, 10 mM MgCl₂, pH7.5. The data were recorded in 5 to 13 videos per dataset, on one or more days. The number of individual molecules included per histogram are: A577I, 154; wt, 163; +Aha1, 366; -Aha1, 231; +crowding, 50; -crowding, 81. All smFRET traces are available from *Figure 2—source data 1*.

The online version of this article includes the following source data and figure supplement(s) for figure 2:

**Source data 1.** smFRETdatasets.zip.
**Figure supplement 1.** Example traces of the reference datasets.
**Figure supplement 2.** Fits and fit coefficients.
**Figure supplement 3.** Single vs double A577I mutation.
**Figure supplement 4.** Macro-molecular vs small molecular crowding.

conformational changes (*Schmid et al., 2016*; *Roy et al., 2008*; *Lerner et al., 2018*; *Mazal and Haran, 2019*). Example traces obtained from three Hsp90 molecules under different conditions are displayed in *Figure 2b*: First for the point mutant A577I, then in the presence of the cochaperone Aha1, and lastly under macro-molecular crowding by Ficoll400. The observed transitions between the low- and high-FRET states reflect global opening or closing. *Figure 2c* shows the steady-state (equilibrium) population of open and closed conformations. In all three cases, a shift toward closed conformations is observed with respect to the corresponding reference distribution obtained under equivalent experimental conditions (see caption of *Figure 2* and Materials and methods for details). The optimal buffer conditions for each of the three cases differ slightly, which explains the differences between the three reference datasets displayed in *Figure 2c* (black lines). Relative populations and their uncertainties were obtained by bootstrapping (see Materials and methods and *Supplementary file 1*).

The A577I mutation increased the closed population from (13 ± 3.5)% to (51 ± 5.5)%. This is a large effect considering that no big change in hydrophobic nor charged interactions is expected to result from this hydrophobic-to-hydrophobic point mutation. In particular, as the slightly bulkier iso-leucine side chain points outward in the crystal structure of Hsp90's closed conformation (*Ali et al., 2006*). In addition to the A577I homodimer, already the A577I/wild type (wt) hetero-dimer shows a considerably larger population of closed conformations, especially in the presence of ATP (*Figure 2—figure supplement 3* (left)). Under ADP conditions the additive effect is also observed, but weaker (*Figure 2—figure supplement 3* (right)). Under both conditions, the second A577I in the homodimer leads to a further shift toward closed conformations. The slight but consistent shift of the corresponding low-FRET peak in *Figure 2c* (left) could be explained by a sterical hindrance of the farthest opening in the A577I homodimer. Furthermore, very fast transitions at the temporal res-olution limit (200 ms) occurred more frequently, which can be seen by the increased overlap between the two populations. Both, sterical hindrance and faster transitions, can be interpreted as a global stiffening of Hsp90's structural core formed by the C- and middle domain. The interaction with Aha1 (*Figure 2c*, center) forms inter-domain (N–M) and inter-monomer contacts. The latter increase the affinity for N-N binding, and thus cause Hsp90 to shift from (24 ± 4.0)% closed to (48 ± 5.0)% closed population, which is in line with previous qualitative reports (*Hessling et al., 2009*; *Wortmann et al., 2017*). Lastly, macro-molecular crowding increased the closed population from (11 ± 4.5)% to (50 ± 10.5)%, in agreement with previous ensemble findings (*Halpin et al., 2016*). In contrast, to the A577I mutant, crowding slowed down fast fluctuation at the resolution limit, which creates well-separated populations in *Figure 2c*). The induced shift toward closed con-formations appears in a concentration-dependent manner, as can be seen in *Figure 2—figure sup-plement 4*. In contrast to polymeric, branched sucrose (Ficoll400), monomeric sucrose had only negligible effect on the steady-state populations. This proves that macro-molecular crowding is the cause of the observed population shift, and a biochemical glucose-associated reason can be dismissed.

In all three cases, a clear shift toward closed conformations is observed, although to slightly dif-ferent extents. Importantly, based on these distributions alone, the *energetic* origin of the popula-tion shift remains unclear. That is whether the observations arise from a stabilization of closed conformations, or a destabilization of the open conformations, or even a combination of both. To answer these questions, we solved the full kinetic rate model, presented below.

## Same thermodynamics, different conformational kinetics

*Figure 3a* shows the kinetic rate models describing Hsp90's global opening and closing dynamics. The significant changes caused by each type of modulation are highlighted in red and green as indi-cated. The corresponding quantitative rate changes and confidence intervals are displayed in *Figure 3b*. We used the Single-Molecule Analysis of Complex Kinetic Sequences (SMACKS [*Schmid et al., 2016*]) to quantify rate constants and uncertainties directly from the smFRET raw data. For Hsp90's global conformational changes, we consistently infer 4-state models (*Schmid et al., 2016*; *Schmid et al., 2018*; *Schmid and Hugel, 2018*): two low-FRET states (open conformations) and two high-FRET states (closed conformations). Although only two different FRET efficiencies can be resolved, at least four kinetic states are needed to describe the observed kinetic heterogeneity. Based on recent results (*Hellenkamp et al., 2017*), we expect an entire ensemble of open sub-conformations that - on the timescale of the experiment - are sufficiently well described by two kinetically different low-FRET states. The two closed states, one short-lived (state 2) and one lon-ger-lived (state 3), likely differ in local conformational elements (*Giannoulis et al., 2020*; *Huang et al., 2019*). The well-known N-terminal beta-sheet with or without its cross-monomer con-tacts (as observed in the closed crystal structure [*Ali et al., 2006*]) could explain the additional stabi-lization of state 3 with respect to state 2. All states 0, 1, 2, 3 of this highly dynamic Hsp90 system represent conformational ensembles that are defined by their FRET efficiency and their kinetics (rather than discrete or 'frozen' conformations).

In the case of the cochaperone Aha1, the shift in the FRET efficiency histogram originates from opposing changes of the fast rates between states 1 and 2 (*Figure 3—figure supplement 1*). This is in contrast to the effect of the C-terminal point mutation A577I, which collectively accelerates the pathway from state 0 via state 1 and state 2 to state 3. Note that a kinetic model with only three links - similar to the model for crowding in *Figure 3a*, right - is statistically sufficient (according to

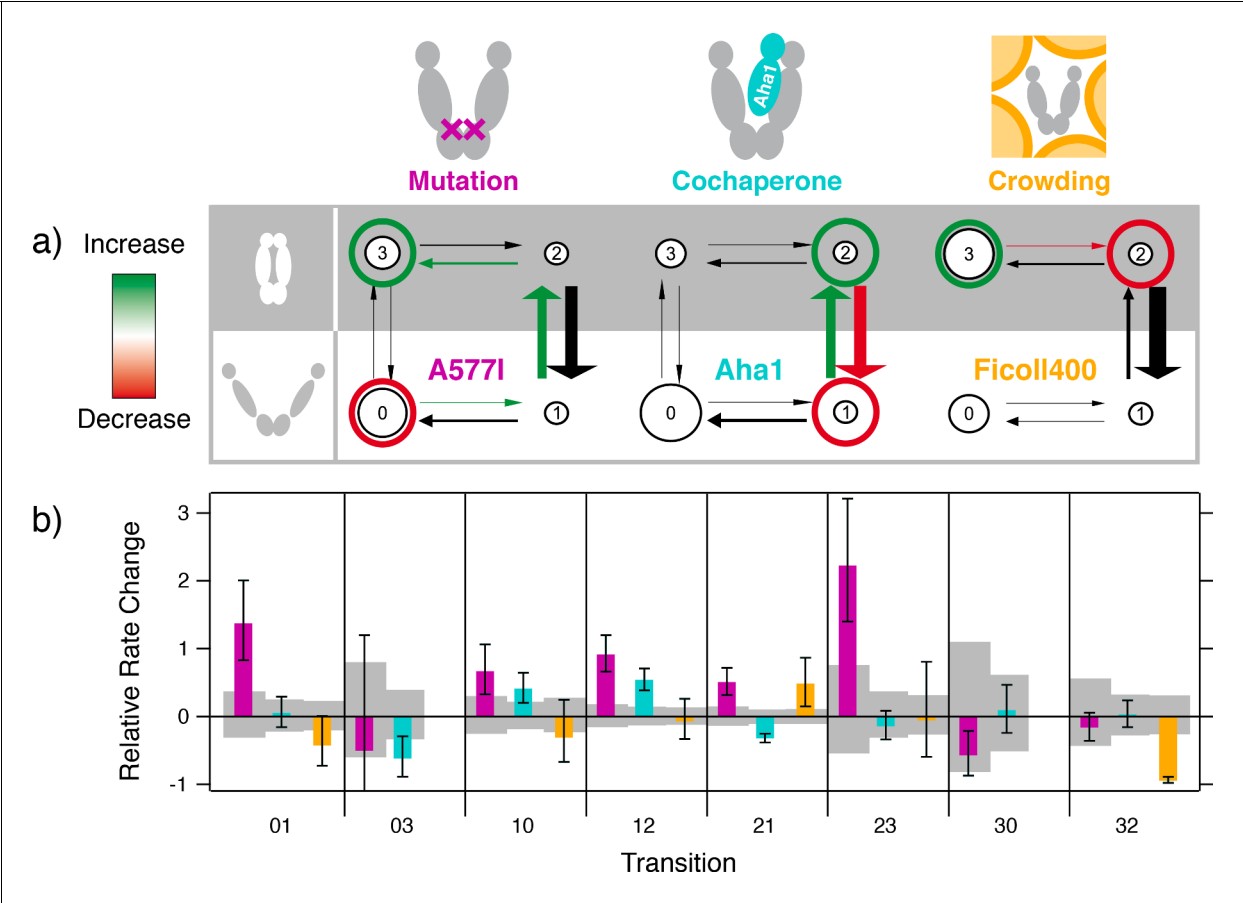

**Figure 3.** Different conformational kinetics cause similar thermodynamics. (**a**) Kinetic rate models observed for the point mutant A577I, the cochaperone Aha1, the macro-molecular crowding agent Ficoll400 - each compared to the reference data: wild-type, no Aha1, no crowding, respectively. Significant differences to the reference are highlighted in red and green. Conformational kinetics are described by four states: states 0,1 represent open conformations, and 2,3 are closed conformations. Large and small arrows and circles indicate the size of rates and populations, respectively. For crowding, only 3 links are found. (**b**) The relative rate change under the three conditions in (**a**) with respect to the reference. Gray boxes show the 95% confidence interval of the reference data. Transition names and color code as in (**a**). All values are listed in ***Supplementary file 1C***. The molecule counts are the same as for ***Figure 2***.

The online version of this article includes the following figure supplement(s) for figure 3:

**Figure supplement 1.** Rate constants and 95% confidence intervals for Hsp90 and Aha1.
**Figure supplement 2.** Rate constants and 95% confidence intervals for Hsp90 and mutations.
**Figure supplement 3.** The effect of crowding on Hsp90's opening and closing transitions.

likelihood ratio testing detailed in ***Schmid et al., 2016*** SI point 1.2) to describe the observed kinetics of the A577I homodimer, implying less kinetically heterogeneous fluctuations (***Figure 3—figure supplement 2***).

Under macro-molecular crowding, changes of the conformational dynamics are visible already from the stationary distributions: as shown in ***Figure 2a***, the low and high FRET populations are most separated in this case. This is indicative of fast fluctuations, at or below the timescale of the sampling rate (5 Hz) that are slowed down at higher viscosity. Still, transitions between open and closed conformations are regularly observed in the experiment (***Figure 3—figure supplement 3***). For the fully resolved kinetics, the main difference is observed for the rates between the closed states 2 and 3. This agrees with the increase of the closed population under macro-molecular – but not small molecular – crowding. Altogether, ***Figure 3*** shows three completely different kinetic effects that underlie seemingly analogous ensemble behavior.

Based on the complete kinetic rate models, we can now deduce the impact on free energies along a specific spatial reaction coordinate, here the N-terminal extension (***Figure 4***). The point

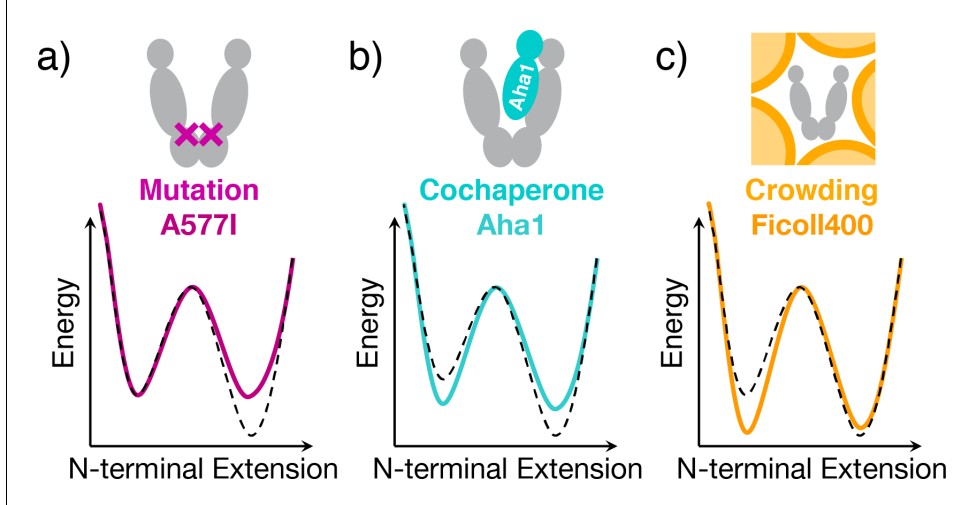

**Figure 4.** Three contrasting effects on Hsp90's conformational energy landscape. (**a**) The open conformation is destabilized by the A577I mutation. (**b**) Aha1 inversely affects both equilibria. (**c**) Macro-molecular crowding stabilizes the closed conformation. The dashed black line indicates the reference. This mechanistic information was obtained from all six rate models represented in *Figure 3*, it is not accessible from *Figure 2c* alone. To illustrate the changes on the global energy landscapes, we chose the transition state as a reference for the measured relative energies.

mutation A577I causes an asymmetric destabilization of open conformations. Aha1 leads to simultaneous destabilization of open conformations and stabilization of closed conformations, whereas macro-molecular crowding only stabilizes closed conformations. Importantly, this information is not accessible from the steady-state distributions in *Figure 2c*.

## ATPase stimulation – to varying degrees

The collective shift toward closed conformations is accompanied by an overall increase in ATPase activity under all three conditions (*Figure 5*): 7-fold for A577I, 17-fold for Aha1, 4-fold for crowding, respectively. Thus, the increase in ATP hydrolysis rate does not reflect the increase in the closed population observed in *Figure 2c*. This is a first indication of causalities that involve more than just the occurrence of closed conformations. In the following, we dissect the molecular origins of the increased ATPase activity. The effect of macro-molecular crowding can serve as an estimate of the ATPase stimulation caused exclusively by the relative stabilization of closed conformations, because a biochemical interaction of sucrose was excluded in control experiments (see above). Remarkably, the entirely non-specific interaction leads already to a considerable ATPase acceleration of a factor 4. This supports the wide-spread notion that the closed conformation represents Hsp90's active state (*Prodromou, 2012*). But, in comparison to the biochemical effect of the cochaperone Aha1, the stimulation by crowding is still modest, despite the much larger closed population. Specifically, the 4.5-fold increased closed population, comes with a 4-fold increased ATPase activity, whereas in the presence of Aha1 already a 2-fold increased closed population is accompanied by a 17-fold ATPase stimulation. The fact that Aha1 induces the smallest increase in closed population, but under the same conditions the largest ATPase stimulation, highlights the functional importance of *specific* contacts between Aha1 and Hsp90, which are responsible for 88% of the ATPase stimulation by Aha1.

In the case of the A577I mutant a 3.9-fold increased closed population comes with a 7-fold ATPase amplification. This could result from the mentioned hindrance of extremely open states indicating a conformational stiffening, restricting Hsp90's native flexibility. In other words, not only changes in the equilibrium, but also changes in the kinetics affect the ATPase activity as discussed below.

A comparison of the observed changes on the conformational steady-state and the ATPase rate allows us to estimate the contribution of conformational confinement in each case

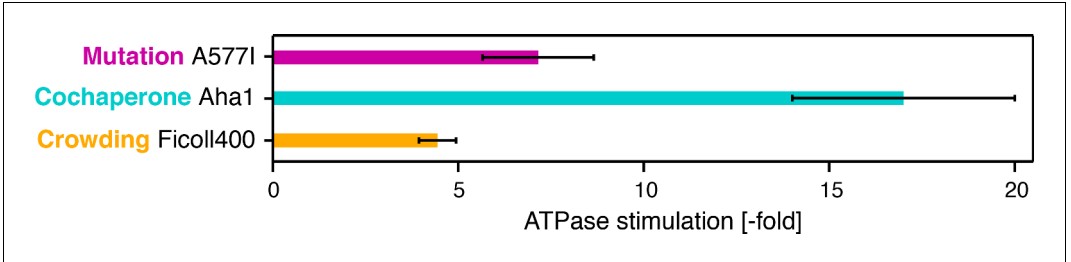

**Figure 5.** ATPase stimulation by local and global modulations. Namely by the point mutation A577I (*Retzlaff et al., 2009*), in the presence of 3 μM Aha1 (*Jahn et al., 2014*) and 20wt% Ficoll400, respectively. The ATPase stimulations are normalized by the activity of the wild type, without Aha1 and without crowding, respectively.

(*Supplementary file 1B*). We find that conformational confinement accounts for 100% of the ATPase stimulation induced by macro-molecular crowding, for 60% of the stimulation by the point mutation A577I, and for only 10% of the stimulation by cochaperone Aha1. The contribution of conformational confinement is thus largest for purely non-specific interactions, while additional allosteric effects may increase the stimulation by A577I, and specific rearrangements of the active site dominate the stimulating action of cochaperone Aha1. Our data do not indicate additional combinatorial effects of these three stimulations, which can be investigated in a future study.

## Discussion

Herein we compare three types of Hsp90 modulations, spanning a wide range from a site-specific point mutation, via cochaperone binding, to completely non-specific macro-molecular crowding. All three modulations provoke a similar steady-state (equilibrium) behavior, namely an increase in Hsp90's closed conformation and in Hsp90's ATPase rate. In contrast, they show significant differences in their kinetics, which could be revealed by single-molecule FRET. This can be rationalized by the emerging picture of yeast Hsp90, as a very flexible dimer that relies critically on external assistance (e.g. by cochaperones) to control its non-productive flexibility. For example, Hsp90's ATPase function requires the concerted action of the N-terminal nucleotide binding pocket with the ATP lid and distant elements such as the catalytic loop of the middle domain and parts of the opposite N-domain (the N-terminal β1-α1 segment). These elements – also called the catalytic unit (*Prodromou, 2012*) - however, are very flexible, such as the entire multi-domain dimer. Consequently, anything that constrains this flexibility and confines Hsp90 in a more compact conformation, has a high potential to increase the combined probability for such a concerted action - be it by specific or even non-specific interaction.

*Figure 6* shows that the functional impact of conformational confinement can be understood as a direct result of combinatorics: in a flexible protein such as Hsp90, the catalytically active elements have many translational and rotational degrees of freedom. Thus, the probability for a certain hydrolysis-competent conformation is very small. It is however increased dramatically by conditions that constrain these degrees of freedom – even non-specifically – and localize the catalytically active elements. This notion can be further extended to cochaperone binding, and it also explains mutual effects upon client interaction. We conclude, while Hsp90's flexibility may facilitate its numerous interactions with diverse clients and cochaperones, the flexibility itself has substantial off-state character regarding the ATPase function of Hsp90. In turn, by directly affecting the lifetime of the nucleotide/active site contacts and the resulting structural stabilization, ATP hydrolysis controls also longer-range allosteric effects linked to these residues.

A closer look at the regulation of Hsp90's conformational energy landscape by single-molecule FRET shows the many ways to reach similar ensemble results. The point mutation A557I destabilizes the open conformation, most probably by preventing access to a subset of the conformational ensemble of open states. This leads to an overall stiffened structure (more confined) of the A577I homodimer and therefore less extensive random walks, that is a more streamlined conformational transition toward the closed conformations. Macro-molecular crowding stabilizes the closed

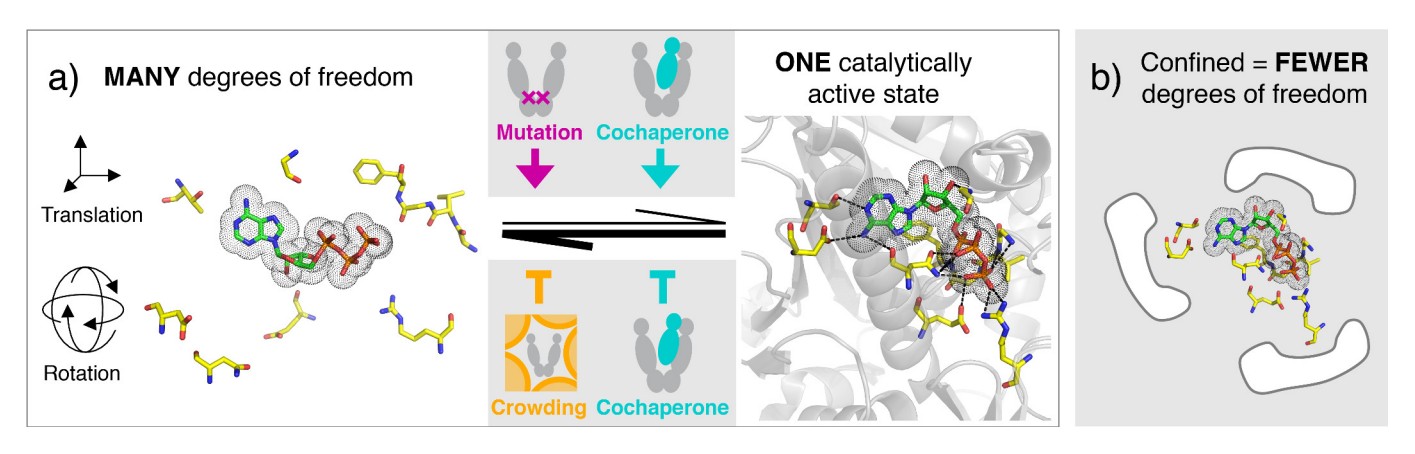

**Figure 6.** Stimulation of protein function by conformational confinement - a result of combinatorics. Enzymatic binding pockets of proteins are often composed of flexible elements with many degrees of freedom, (a) left. All of these have to adopt a specific 3D configuration to form the catalytically active state, (a) right. For example in Hsp90, the hydrolysis competent state includes direct contacts with residues from different domains: G100, T171, D79, N37, E33, R380, and G121-F124 (counter-clockwise from top, in the left panel). As a result, the probability to be in the catalytically active state is very small, compared to the combined probability for all other accessible conformations (left), and the catalytic activity is low. Intriguingly, specific or non-specific confinement reduces the non-productive degrees of freedom in Hsp90, leading to a relative stabilization of the hydrolysis-competent state and therefore stimulation of the catalytic activity, as illustrated in (b). Here we have shown that this can occur through allosteric effects of mutations, protein-protein interaction, or macromolecular crowding. Structures were visualized with Pymol, based on pdb entry 2cg9.

conformation by simple, sterical confinement. The cochaperone Aha1 combines both mechanisms with additional specific rearrangements. In theory, all three modulations can lead to the exactly same thermodynamic observation, and in fact we observe very similar steady-state distributions. Nevertheless, the kinetics may still vary significantly – as experimentally demonstrated herein. This is an important mathematical fact that holds true for all protein systems.

Moreover, our findings indicate that regulation by cochaperones - and protein-protein interactions in general - can have far-reaching thermodynamic, kinetic, and functional consequences. Some of them can possibly be mimicked by individual point mutations, but other consequences might be missed. Lastly, as shown in this work, already non-specific, purely physical macro-molecular crowding has strong effects on thermodynamics, kinetics and function, therefore caution is advised when relating in vitro findings to in vivo function. Certain mutations that show little or no ATPase activity in vitro, may be 'rescued' by the conformational confinement in the crowded cellular environment, and still be ATPase active in vivo. As demonstrated herein, in many in vitro experiments macro-molecular crowding could easily be included.

In conclusion, we demonstrated herein three ways of protein regulation, ranging from site-specific localized to global modulations. All three ways show very similar thermodynamic observations, which are, however, caused by clearly different conformational kinetics. This is direct evidence for the importance of kinetics, and of the *dynamic* structure–function relationship in proteins. The reduction of non-productive structural flexibility stimulates Hsp90's ATPase function even by entirely non-specific means. Our findings demonstrate that functional stimulation as a result of conformational combinatorics plays an important role in protein regulation. We anticipate that such conformational confinement - by localized or global modulations - is an important mechanistic concept with widespread implications for protein function in diverse systems.

## Materials and methods

### Protein construct preparation

Yeast Hsp90 dimers (UniProtKB: P02829) with a C-terminal coiled-coil motif (kinesin neck region of *D. melanogaster*) were used to avoid dissociation at low concentrations (*Mickler et al., 2009*). Previously published cysteine positions (*Hessling et al., 2009*) allowed for specific labeling with donor

(61C) or acceptor (385C) fluorophores (see below). Point mutation A577I was introduced using Quik-Change Lightning Site-Directed Mutagenesis Kit (Agilent Technologies). The constructs were cloned into a pET28b vector (Novagen, Merck Biosciences, Billerica, MA). They include an N-terminal His-tag followed by a SUMO-domain for later tag cleavage. The QuickChange Lightning kit (Agilent Technologies, Santa Clara, CA) was used to insert an Avitag for specific *in vivo* biotinylation at the C-terminus of the acceptor construct. *Escherichia coli* BL21star cells (Invitrogen, Carlsbad, CA) were cotransformed with pET28b and pBirAcm (Avidity Nanomedicines, La Jolla, CA) by electropora-tion (Peqlab, Erlangen, Germany) and expressed according to Avidity's *in vivo* biotinylation proto-col. The donor construct was expressed in *E. coli* BL21(DE3)cod+ (Stratagene, San Diego, CA) for 3 hr at 37°C after induction with 1 mM isopropyl β-D-1-thiogalactopyranoside (IPTG) at $OD_{600} = 0.7$ in $LB_{Kana}$. A cell disruptor (Constant Systems, Daventry, United Kingdom) was used for lysis in both cases. Proteins were purified as published (*Jahn et al., 2014*) (Ni-NTA, tag cleavage, anion exchange, size exclusion chromatography). 95% purity was confirmed by SDS-PAGE. Fluorescent labels (Atto550- and Atto647N-maleimide) were purchased from Atto-tec (Siegen, Germany) and coupled to cysteins according to the supplied protocol. If not stated differently, all chemicals were purchased from Sigma Aldrich, St. Louis, MO.

Single-molecule FRET (smFRET) was measured as previously detailed using a home built TIRF setup (*Schmid et al., 2016*). Hetero-dimers (acceptor + donor) were obtained by 20 min incubation of 1 μM donor and 0.1 μM biotinylated acceptor homodimers in measurement buffer (40 mM Hepes, 150 mM KCl, and 10 mM $MgCl_2$, pH7.5) at 47°C. This favors biotinylated heterodimers to bind to the polyethylene glycol (PEG, Rapp Polymere, Tuebingen, Germany) passivated and neutravidin (Thermo Fisher Scientific, Waltham, MA) coated fluid chamber. Residual homodimers are recognized using alternating laser excitation (ALEX) of donor and acceptor dyes (*Lee et al., 2005*; *Hellenkamp et al., 2018*) and excluded from analysis. For optimal interaction affinity with Aha1, measurements were performed in low salt buffer (40 mM Hepes, 20 mM KCl, 5 mM $MgCl_2$, pH 7.5 with 3.5 μM Aha1 and 2 mM ATP). For comparison, data without Aha1 were measured accordingly. Notably, significant binding was previously found for Aha1 with labeled Hsp90-385C at much lower concentration of 0.3 μM (*Li et al., 2013*), which is exactly the dissociation constant reported for unlabeled Hsp90 (*Panaretou et al., 2002*). This implies that, although not directly detectable in the experiment, Hsp90 exists predominantly in complex with Aha1 under the used conditions. A577I/wt constructs were created through monomer exchange (see above). They are distinguished from both kinds of homodimers through the fluorescence signal (donor+acceptor). Measurements were performed in measurement buffer plus 2 mM ATP if not stated differently. Macro-molecular crowding was mim-icked by 20wt% polymeric sucrose, known as Ficoll400 (Sigma Aldrich) in measurement buffer if not stated differently.

## Single-Molecule FRET data analysis

Single-molecule FRET trajectories were selected, and corrected for background, donor leakage, acceptor direct excitation, as well as different fluorescence quantum yields, detection efficiencies, excitation efficiencies, and laser powers, as described before (*Hellenkamp et al., 2018*).

The uncertainties of steady-state populations were estimated by bootstrapping: out of each data-set, random subsets (or samples) containing two thirds of the total number of traces were drawn with replacement. A histogram was compiled for each subset, and fitted with a double-Gaussian fit function and fixed peak positions from *Figure 3—figure supplement 2*. Relative populations were obtained for each histogram. This was repeated for 1000 subsets. We report the means of 1000 such populations, and as an uncertainty measure their standard deviations (rounded upwards to 0.5). See also *Supplementary file 1A*.

SMACKS is based on two-dimensional semi-ensemble Hidden Markov models (HMMs), as described in detail earlier (*Schmid et al., 2016*). In brief, smFRET donor and acceptor fluorescence traces serve as input data. SMACKS comes in two steps. First, individual HMMs are optimized for each molecule separately, to allow for the experimentally observed intensity differences between dif-ferent molecules. The resulting emission parameters for each molecule – that is Gaussian mean and variance representing the fluorescence intensity levels - are then used for the second step: the semi-ensemble HMM optimization. Now the kinetic parameters – so-called start and transition probabili-ties – are ensemble-optimized based on an entire dataset, using the individual emission parameters of the first step. The transition probabilities are converted into rate constants by considering the

sampling rate of the experiment (frames per second). Several such HMM optimizations with increasing model size (i.e. numbers of states) are calculated, and compared using the Bayesian Information Criterion (BIC), which reproducibly yields four states for Hsp90's characteristic conformational dynamics: two open, two closed. Within the four-state model, we determine the most likely number of links using likelihood ratio tests following the hierarchical search for simplified models described in *Bruno et al., 2005*.

## Activity assay

ATPase activity was measured at 37°C coupled to NADH oxidation, which was followed as a decrease in absorption at 340 nm using an ATP regenerative assay similar to *Tamura and Gellert, 1990*: 0.2 mM NADH Di-Na (Roche, Basel, CH); 2 mM phosphoenol pyruvate K-salt (Bachem, Bubendorf, CH); 2 U/ml pyruvate kinase, Roche; 10 U/ml lactate dehydrogenase, Roche; in 40 mM Hepes, 150 mM KCl, 10 mM $MgCl_2$, pH 7.5.

## Acknowledgements

The cochaperone Aha1 was a kind gift of Dr. Markus Jahn. We thank Dr. Markus Götz for previous contributions to data analysis, and Dr. Eli van der Sluis for helpful comments on the manuscript. This work has been funded in part by the European Research Council through ERC grant agreement no. 681891 and the Deutsche Forschungsgemeinschaft (DFG, German Research Foundation) – Project-ID 403222702 – SFB 1381. SS acknowledges the Postdoc.Mobility fellowship no. P400PB_180889 by the Swiss National Science Foundation.

## Additional information

### Funding

| Funder | Grant reference number | Author |
|---|---|---|
| European Commission | 681891 | Thorsten Hugel |
| Deutsche Forschungsgemeinschaft | SFB 1381 | Thorsten Hugel |
| Swiss National Science Foundation | Postdoc.Mobility fellowships | Sonja Schmid |

The funders had no role in study design, data collection and interpretation, or the decision to submit the work for publication.

### Author contributions

Sonja Schmid, Conceptualization, Data curation, Software, Formal analysis, Investigation, Visualization, Methodology, Writing - original draft, Writing - review and editing; Thorsten Hugel, Conceptualization, Supervision, Funding acquisition, Validation, Visualization, Project administration, Writing - review and editing

### Author ORCIDs

Sonja Schmid https://orcid.org/0000-0002-3710-5602
Thorsten Hugel https://orcid.org/0000-0003-3292-4569

### Decision letter and Author response

Decision letter https://doi.org/10.7554/eLife.57180.sa1
Author response https://doi.org/10.7554/eLife.57180.sa2

## Additional files

### Supplementary files

• Supplementary file 1. Supplementary tables. (**A**) The percentage of open and closed populations and their standard deviation. (**B**) The ATPase stimulation factors and conformational confinement factors. (**C**) The transition rates and their confidence intervals.

• Transparent reporting form

### Data availability

All single-molecule FRET traces for each dataset are provided as source data (smFRETdatasets.zip). The source code of SMACKS is available for download here: https://www.singlemolecule.uni-freiburg.de/SMACKS.

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
