## [Decision Letter]

Thank you for submitting your article "Controlling protein function across scales – from point mutation to crowding" for consideration by *eLife*. Your article has been reviewed by three peer reviewers, and the evaluation has been overseen by a Reviewing Editor and José Faraldo-Gómez as the Senior Editor. The following individuals involved in review of your submission have agreed to reveal their identity: Michael Schlierf (Reviewer #2); Timothy Craggs (Reviewer #3).

The reviewers have discussed the reviews with one another and the Reviewing Editor has drafted this decision to help you prepare a revised submission.

We would like to draw your attention to changes in our revision policy that we have made in response to COVID-19 (https://elifesciences.org/articles/57162). Specifically, when editors judge that a submitted work as a whole belongs in *eLife* but that some conclusions require a modest amount of additional new data, as they do with your paper, we are asking that the manuscript be revised to either limit claims to those supported by data in hand, or to explicitly state that the relevant conclusions require additional supporting data. In the case of your paper, the manuscript can benefit from some combinatorial experiments (see suggestions below).

Summary:

Schmid and Hugel present a detailed smFRET study on HSP90 conformational kinetics as it transitions between four conformational states. The authors investigate the effects of mutations, co-chaperones and crowding on state population and kinetics. What add significance to the study is a demonstration of the effects of molecular crowding, which is used to simulate conditions resembling cellular environment. Most prominent finding here is a demonstration that variations in experimental conditions (e.g. mutations or crowding) may have similar thermodynamic effects as revealed via population experiments, while very different kinetic effects. Thus, the study highlight the importance to discriminate steady-state thermodynamic outputs from kinetic effects and the ability of smFRET approach to extract both parameters using the same set of data.

All reviewers and the editor agree that the study is important and well executed. The results are interesting and novel, and the concepts discussed are applicable to many conformationally dynamic protein systems.

Essential revisions:

While the reviewers agree that there is no need for essential additional data, we felt that if you have some combinatorial experiments, e.g. Aha1 and crowding to support their claims of combinatorial effects (Figure 6). Otherwise, you could weaken some of the concluding claims.

Please address the following questions regarding the interpretation and data analysis:

1) Figure 2 and data therein:

Having the conditions for the three different experiments in the legend would be helpful. Does this explain the difference in the reference data (i.e. the ionic strength effects the overall conformational profile)? It would be useful to provide some quantitation of the mid-point of the FRET populations (or present them in a vertical stack to aid comparison.) Are the reference data for Figure 2C plots 1 and 3 measured under identical conditions? If so, is the difference between these data significant?

2) In Figure 2B, please show representative traces for WT sample without cochaperone or crowding factor; change the format of all the traces so that the idealization quality can be checked visually by reader. The fitting of time traces to determine the kinetics of transitions is central to the presented work. Please provide some detail as to how the SMACKS algorithm works. Does it involve a global fit to ALL the data? From the results presented, it is not clear how you rule out transitions between, for example state 0 and state 2. Is this just what falls out of the data analysis?

3) 4 states are proposed to fit the kinetic model best. You assume that HSP90 is adapting the same identical molecular conformations, meaning state 0, 1, 2, and 3 are the same states in presence or absence of e.g. the co-chaperone. However, the peaks in the FRET efficiency histograms, especially of the high FRET efficiency state, but also the low FRET efficiency state (in A577I) are strongly shifted in presence of all modifications, which indicate a slightly different molecular conformation. For a strict comparison of free energy landscapes as suggested in Figure 4 no common reference state is therefore available. Please discuss whether the kinetics states are necessarily different conformations to which your current signal is blind?

4) As a common reference state, the transition state between the N-terminal compact and N-terminal expanded state id used (Figure 4). Is there any evidence that this is possible, why not the compact or expanded state?

5) Figure 4: From your kinetics you have some idea of the rates of the transitions between the 2 open conformations and the two closed conformations which you could show in a supplementary figure? This is potentially helpful because for example Figure 4C it is really state 3 that is stabilized with respect to state 2.

6) Each measured reference should be identical. Yet, the population of the high FRET efficiency state varies from 14 to 29%. This is a two-fold difference, which is also what kinetically is named as a significant difference in transition rates. Can you explain this difference, explain how the state occupancies in the subsection “Mutation, cochaperone and crowding show similar thermodynamics” have been determined and what the relative error is? It would further be good to also show example traces of the reference experiments with wt in the figure supplements to give the reader a chance for comparison beyond the state histogram and rate summaries.

7) Please expand the following discussion: What does the current work allow the authors to conclude about the function of the ATPase activity in HSP90? Especially delineate the physico-chemical conclusions from the functional conclusion.

8) Figure 6 suggests combinatorial effects. By extension, this suggests that crowding and the point-mutation would have an even more enhanced effects of transition rates or ATPase activity. Can you show evidence by a combination of any two of their conditions and support the combinatorial suggestion from Figure 6?

9) A clearer description of the data analysis needs to be placed in the Materials and methods section.

10) Statistical errors are reported for the interconversion rates between the various conformational states detected. Please also specify whether the results of each experimental condition were obtained from a single experiment (i.e. video) recorded in a single day, or that the results of each experimental condition were obtained from a set of videos recorded on different days to check for reproducibility? Regarding the state populations an error of the extracted percentages would be helpful. This can be done e.g. by bootstrapping.

---

## [Author Response]

Essential revisions:While the reviewers agree that there is no need for essential additional data, we felt that if you have some combinatorial experiments, e.g. Aha1 and crowding to support their claims of combinatorial effects (Figure 6). Otherwise, you could weaken some of the concluding claims.

Thank you for pointing out that Figure 6 could be misunderstood as claiming combinatorial effects, e.g. of Aha1 and crowding. This was not our intention: please note the ‘or’ in the legend of Figure 6: “Intriguingly, specific *or* non-specific confinement reduces the non-productive degrees of freedom in Hsp90,…” and also: ”Here we have shown that this can occur through allosteric effects of mutations, protein-protein interaction, *or* macromolecular crowding.”

Since this is indeed an important point, we added the following text to clarify that herein we do *not* claim combinatorial effects of point mutation, cochaperone interaction, and macro-molecular crowding. In contrast, we describe how conformational confinement plays a role in all three cases:

“A comparison of the observed changes on the conformational steady-state and the ATPase rate allows us to estimate the contribution of conformational confinement in each case (see Table 2 in Supplementary file 1). […] Our current data does not indicate additional combinatorial effects of these three stimulations, which can be investigated in a future study.”

Lastly, we found a mistake in Figure 6, which depicted the representation of the point mutation A577I in place of the co-chaperone Aha1 – and vice versa. We corrected this mistake, such that Figure 6 is now in line with all the data, and the text.

Please address the following questions regarding the interpretation and data analysis:1) Figure 2 and data therein:Having the conditions for the three different experiments in the legend would be helpful.

We thank the reviewers for their suggestion, and added the experimental conditions now also to the caption of Figure 2, which will help the reader to find them.

Does this explain the difference in the reference data (i.e. the ionic strength effects the overall conformational profile)?

Yes, the different measurement conditions explain the differences between the three reference datasets. E.g. the lower salt concentration that is necessary for optimal Aha1 binding, leads to a slightly larger high-FRET (closed) population (as reported previously in Schmid and Hugel, 2018; Figure 2C). We added the following sentence to the main text to clarify this:

“The optimal buffer conditions for each of the three cases differ slightly, which explains the differences between the three reference datasets displayed in Figure 2C (black lines).”

It would be useful to provide some quantitation of the mid-point of the FRET populations (or present them in a vertical stack to aid comparison.)

We thank the reviewers for this suggestion. We present the panels vertically now, to better visualize the mid-points of the FRET population. In addition, we added a new Figure 2—figure supplement 2 showing all double-Gaussian fits and their fit coefficients.

Are the reference data for Figure 2C plots 1 and 3 measured under identical conditions? If so, is the difference between these data significant?

No, Figure 2C plot 1 includes 2 mM ATP, while plot 3 does not, as described above. We thank the reviewers for pointing out that this was not clear enough. This important information is now not only found in the Materials and methods section, but also in the legend of Figure 2.

2) In Figure 2B, please show representative traces for WT sample without cochaperone or crowding factor.

We added a new Figure 2—figure supplement 1 with representative traces of all three reference datasets.

Change the format of all the traces so that the idealization quality can be checked visually by reader.

We are not sure what the reviewers mean by ‘idealization quality’. It could mean a likelihood score for each datapoint’s assignment to a certain state (0 to 3). The γ probabilities (after Rabbiner’s nomenclature for Hidden-Markov models; Rabiner (1989) Proc IEEE) could provide a measure for this. However:

i) SMACKS does currently not output this information. We agree such a feature could give a relative rating of state assignment reliability per datapoint, and we gratefully keep this suggestion in mind for future versions of SMACKS.

ii) It is not the standard in the field: we know exactly one smFRET kinetics publication, which does report such ‘idealization quality’ albeit for a single-trace analysis: Greenfeld et al. (2012) Plos ONE. Recent smFRET kinetics papers do not report such a quantity, e.g.: Barth,.…, Lamb (2020) JACS;

Fitzgerald,.…, Blanchard (2019) Nature; Zhou,.…, Ha (2019) Nat Chem Biol; Schärfen, Schlierf (2019) Methods; Dulin,.…, Kapanidis (2018) Nat Comm; Kilic,.…, Seidel, Fierz (2018) Nat Comm; van de Meent,.…, Gonzalez (2014) Biophys J.

iii) The uncertainty measures for the kinetic analysis are already provided in the form of confidence intervals for each rate constant specified in Table 1 in Supplementary file 1.

The fitting of time traces to determine the kinetics of transitions is central to the presented work. Please provide some detail as to how the SMACKS algorithm works.

We agree that adding some more detail improves the readability of this manuscript, although SMACKS has been described before in great detail (Schmid, Götz and Hugel, 2016). We added the following paragraph to the Materials and methods section:

“SMACKS is based on two-dimensional semi-ensemble HMM, as described in detail earlier [Schmid, Götz and Hugel, 2016]. In brief, smFRET donor and acceptor fluorescence traces serve as input data. ][…] Within the four-state model, we determine the most likely number of links using likelihood ratio tests following the hierarchical search for simplified models described in [Bruno et al., 2005].”

Does it involve a global fit to ALL the data?

Yes, SMACKS involves a global HMM optimization of the transition probabilities based on all molecules in a given dataset (however *not* on all six datasets at once).

From the results presented, it is not clear how you rule out transitions between, for example state 0 and state 2. Is this just what falls out of the data analysis?

For two open (o), and two closed (c) states, the maximum number of mathematically identifiable links is four (eight rate constants) – independent of data quality. For four links, there are two mathematically equivalent cyclic models (-o-o-c-c- or -o-c-o-c-); in the case of three links, there are three mathematically equivalent models, as discussed in Schmid and Hugel, 2018, in Figure 5A, and previously for ion channel data by Bruno et al., 2005. To select one of the mathematically equivalent models, we used prior knowledge about Hsp90’s structure, which led to the presented models with the fewest open-to-closed links, for three reasons:

i) The long-distance transition path of ca. 4nm between the observed open and closed conformations renders the lowest number of open to close transitions most likely.

ii) From confocal smFRET results (Hellenkamp et al., 2018), we know that the two open states represent a much larger conformational ensemble with shallow energy barriers. Hence, transitions within the conformational ensemble of open states are to be expected.

iii) The long-lived / short-lived closed states are likely the result of added / missing cross-monomer contacts of Hsp90’s most N-terminal β-sheet, which are resolved in the crystal structure by Ali et al., 2006, pdb:2cg9.

Altogether, we dismiss ‘cross transitions’ (0 <-> 2, 1 <-> 3) based on both statistical criteria, and prior knowledge on Hsp90.

3) 4 states are proposed to fit the kinetic model best. You assume that HSP90 is adapting the same identical molecular conformations, meaning state 0, 1, 2, and 3 are the same states in presence or absence of e.g. the co-chaperone. However, the peaks in the FRET efficiency histograms, especially of the high FRET efficiency state, but also the low FRET efficiency state (in A577I) are strongly shifted in presence of all modifications, which indicate a slightly different molecular conformation. For a strict comparison of free energy landscapes as suggested in Figure 4 no common reference state is therefore available. Please discuss whether the kinetics states are necessarily different conformations to which your current signal is blind?

The reviewers are right: there can be small structural differences between states 0,1,2,3. However, such a ‘strict comparison of free energy landscapes’ as mentioned by the reviewers would require that all atom positions are known. This is not feasible experimentally, and zero uncertainty is even theoretically impossible. Therefore, state definitions are used in order to interpret experimental data. This is particularly important to gain insight into dynamic systems, such as Hsp90, where the energy landscape shows many shallow minima leading to conformational ensembles. It is a unique capability of smFRET trajectories that conformational dynamics can be resolved experimentally along a specific reaction coordinate and as a function of time, i.e. not only as fluctuations without direction. We set here the reaction coordinate along the most characteristic open-close transition of Hsp90. And as in all smFRET studies this means that we cannot observe dynamics that are orthogonal to that coordinate. But we *can* rule out the appearance of a new third FRET population that would represent a qualitatively new state. In fact, under all conditions, we separately and consistently find two FRET efficiency populations that split into four kinetic states (whose structural interpretation is described in the subsection “Same thermodynamics, different conformational kinetics”), which makes this global definition of four states the most reasonable one – despite slight shifts of mean positions on a locally shallow energy landscape. Please, note also that small shifts in FRET efficiency arise in dynamic systems already through changes of the kinetics on the order of the time resolution of the experiment.

In summary, we do not claim that states 0,1,2,3 are discrete nor 100% ‘identical molecular conformations’. Instead, they are conformational ensembles defined by their FRET efficiency and their kinetics, which allows us to compare their relative energy levels.

We added the following sentence to the main text:

“All states 0, 1, 2, 3 of this highly dynamic Hsp90 system represent conformational ensembles that are defined by their FRET efficiency and their kinetics (rather than discrete or ‘frozen’ conformations).”

4) As a common reference state, the transition state between the N-terminal compact and N-terminal expanded state id used (Figure 4). Is there any evidence that this is possible, why not the compact or expanded state?

The reviewers are right, only relative energies can be determined. However, if we consider e.g. the case of macro-molecular crowding, the main difference in the observed kinetics leads to a reduced off-rate out of the closed conformations towards the open conformations. In terms of energy landscapes, one can realize this change either by raising two energy levels (the transition state and the global open minimum), or by just lowering one state (the global closed minimum). In line with parsimony arguments (Okham’s razor), we deemed the latter more likely. Analogous arguments apply for the point mutation A577I. Lastly, in case of the binary effect observed with the cochaperone Aha1, we chose to keep the transition state as the reference, for consistency. Importantly, this reference choice does not affect the relative energies nor our conclusions.

We added the following sentence to the legend of Figure 4 to clarify this point:

“To illustrate the changes on the global energy landscapes, we chose the transition state as a reference for the measured relative energies.”

5) Figure 4: From your kinetics you have some idea of the rates of the transitions between the 2 open conformations and the two closed conformations which you could show in a supplementary figure? This is potentially helpful because for example Figure 4C it is really state 3 that is stabilized with respect to state 2.

The reviewers are right, we do resolve the transition rates between the 2 open conformations and the two closed conformations. They are already displayed in Figure 3 along with all other rate constants. Also the absolute values of these rate constants and corresponding confidence intervals are provided in Table 1 in Supplementary file 1.

Lastly, the steady-state changes are shown in the histograms in Figure 2 and population sizes are specified in the main text, and in Table 1 in Supplementary file 1.

The reviewers are also right that under macro-molecular crowding, the kinetic state 3 is stabilized with respect to state 2, which is shown in Figure 3. This results in a smaller open population, since state 2 is the reservoir for any opening transition. The data are fully self-consistent in this way, and no net fluxes or net particle rates can arise in the crowding experiment, which is an equilibrium experiment.

6) Each measured reference should be identical.

The three reference datasets are *not* expected to be identical, as explained under point 1). We are sorry for the confusion.

Yet, the population of the high FRET efficiency state varies from 14 to 29%. This is a two-fold difference, which is also what kinetically is named as a significant difference in transition rates. Can you explain this difference, explain how the state occupancies in the subsection “Mutation, cochaperone and crowding show similar thermodynamics” have been determined and what the relative error is?

Yes, we can explain this difference by the different experimental conditions explained under point 1). The state occupancies (or population sizes) were determined using double-Gaussian fits. They agree very well with the state occupancy determined by SMACKS. We performed additional bootstrapping to estimate the uncertainties of the population sizes, and report now the bootstrapped means and standard deviations both in the main text, and in the new Table 1 in Supplementary file 1. The following description of the bootstrapping procedure was added to the Materials and methods section:

“The uncertainties of steady-state populations were estimated by bootstrapping: out of each dataset, random subsets (or samples) containing two thirds of the total number of traces were drawn with replacement. […] We report the standard deviation of 1000 such populations as an uncertainty measure.”

It would further be good to also show example traces of the reference experiments with wt in the figure supplements to give the reader a chance for comparison beyond the state histogram and rate summaries.

We agree and added the example traces to the new Figure 2—figure supplement 1.

7) Please expand the following discussion: What does the current work allow the authors to conclude about the function of the ATPase activity in HSP90? Especially delineate the physico-chemical conclusions from the functional conclusion.

Our work demonstrates that conformational confinement stimulates the ATPase activity of highly dynamic Hsp90. We link this to the fact that the active site consists of several flexible residues that have to be in a specific arrangement to form the ATPase competent state (Figure 6). Concerning the ATPase activity, we conclude in turn that by directly affecting the lifetime of these contacts and the resulting structural stabilization, ATP hydrolysis itself controls also longer-range allosteric effects linked to these residues.

Moreover, we conclude that Hsp90’s ATPase activity in vivo differs from the typically reported in vitro rate constants. Also, certain mutations (e.g. mimicking PTMs) that show little to no ATPase activity in vitro, may be ‘rescued’ by the conformational confinement in the crowded cellular environment, and still be ATPase active in vivo. This can be tested by ATPase activity assays performed under macromolecular crowding.

We added the following two sentences to the Discussion section:

“In turn, by directly affecting the lifetime of active site contacts and the resulting structural stabilization, ATP hydrolysis itself controls also longer-range allosteric effects linked to these residues.”

“Certain mutations that show little or no ATPase activity in vitro, may be ‘rescued’ by the conformational confinement in the crowded cellular environment, and still be ATPase active in vivo.”

8) Figure 6 suggests combinatorial effects. By extension, this suggests that crowding and the point-mutation would have an even more enhanced effects of transition rates or ATPase activity. Can you show evidence by a combination of any two of their conditions and support the combinatorial suggestion from Figure 6?

We did not study, nor do we claim combinatorial effects as discussed in our first response above.

9) A clearer description of the data analysis needs to be placed in the Materials and methods section.

We thank the reviewers for their suggestion, and have added an additional paragraph to the Materials and methods section. (See main point 2.)

10) Statistical errors are reported for the interconversion rates between the various conformational states detected. Please also specify whether the results of each experimental condition were obtained from a single experiment (i.e. video) recorded in a single day, or that the results of each experimental condition were obtained from a set of videos recorded on different days to check for reproducibility?

We added the following sentence to the caption of Figure 2:

“The data was recorded in 5 to 13 videos per dataset on one or more days.”

Regarding the state populations an error of the extracted percentages would be helpful. This can be done e.g. by bootstrapping.

We agree with the reviewers, and report now mean populations and standard deviations obtained by bootstrapping, both in the main text, and in Table 1 in Supplementary file 1. (See also main point 6.)